# Class-Specific Effects of ARBs Versus ACE Inhibitors on Survival and Cardiovascular Outcomes in MASLD

**DOI:** 10.3390/ijms262010061

**Published:** 2025-10-16

**Authors:** Tom Ryu, Yeon Joo Seo, Jaejun Lee, Ji Won Han, Hyun Yang, Keungmo Yang

**Affiliations:** 1Department of Internal Medicine, Institute for Digestive Research, Digestive Disease Center, College of Medicine, Soonchunhyang University, Seoul 04401, Republic of Korea; tomryu1@schmc.ac.kr; 2Department of Internal Medicine, Division of Gastroenterology and Hepatology, College of Medicine, The Catholic University of Korea, Seoul 06591, Republic of Korea; yj.seo.md@gmail.com (Y.J.S.); pwln0516@gmail.com (J.L.); tmznjf@catholic.ac.kr (J.W.H.); oneggu@naver.com (H.Y.)

**Keywords:** metabolic dysfunction-associated steatotic liver disease, angiotensin II receptor blockers, angiotensin converting enzyme inhibitors, survival, cardiovascular event

## Abstract

Renin–angiotensin system (RAS) inhibitors, including angiotensin-converting enzyme inhibitors (ACEIs) and angiotensin II receptor blockers (ARBs), have been associated with improved outcomes in metabolic dysfunction-associated steatotic liver disease (MASLD). We aimed to assess the differential impact of ACEIs versus ARBs on survival and cardiovascular outcomes in individuals with MASLD. Using data from the UK Biobank, we identified 52,143 participants with exclusive use of either an ACEI or ARB. Individuals with viral, autoimmune, cholestatic, or alcohol-related liver disease were excluded. MASLD was defined as fatty liver index ≥ 60 with ≥1 cardiometabolic risk factor. Inverse probability of treatment weighting (IPTW) was used to adjust for confounders. Outcomes included all-cause mortality, cardiovascular events, hepatic decompensation, and hepatocellular carcinoma (HCC), analyzed using Cox proportional hazards models. Among MASLD participants, ARB use was associated with significantly lower all-cause mortality compared to ACEI use (HR, 0.94; 95% CI, 0.90–1.00; *p* = 0.031) after IPTW adjustment. Cardiovascular risk was also lower with ARBs (HR, 0.92; 95% CI, 0.89–0.96; *p* < 0.001), particularly in subgroups with BMI ≥ 25 kg/m^2^, no diabetes, and advanced fibrosis. No differences in hepatic decompensation or HCC incidence were observed. Benefits of ARBs were not significant in participants without steatotic liver disease. ARB use was associated with improved survival and reduced cardiovascular events in individuals with MASLD, whereas ACEIs expressed no comparable benefit. These findings suggest that ARBs might be a more effective RAS inhibitor subclass in MASLD and support their preferential use in patients with steatotic liver disease requiring antihypertensive therapy.

## 1. Introduction

Metabolic dysfunction-associated steatotic liver disease (MASLD), formerly known as non-alcoholic fatty liver disease (NAFLD), has become the most common cause of chronic liver disease globally, affecting approximately 30% of adults [1,2]. MASLD encompasses a broad clinical spectrum ranging from steatosis to steatohepatitis, fibrosis, cirrhosis, and hepatocellular carcinoma (HCC) [3]. Beyond liver-related outcomes, MASLD is strongly associated with increased risk of cardiovascular disease, which remains the leading cause of mortality in this population [4].

Given the metabolic and inflammatory features of MASLD, there is increasing interest in repurposing existing medications that target key pathogenic pathways [5,6,7]. Among these, renin–angiotensin system (RAS) inhibitors, comprising angiotensin-converting enzyme inhibitors (ACEIs) and angiotensin II receptor blockers (ARBs), have gained attention for their potential hepatoprotective effects [8,9]. Mechanistically, RAS activation promotes hepatic stellate cell activation, oxidative stress, and pro-fibrotic cytokine production, all of which contribute to liver fibrosis progression [10,11]. Several preclinical and clinical studies have demonstrated that RAS blockade attenuates hepatic fibrosis and might reduce portal pressure and inflammation [12,13,14].

Recent observational studies have suggested that RAS inhibitors would improve liver-related and cardiovascular outcomes in patients with MASLD [15,16]. For example, a large multi-center cohort study using real-world data showed that ACEI/ARB use was associated with reduced all-cause mortality, major adverse liver outcomes, and major cardiovascular events when compared to calcium channel blockers [16]. However, these studies have generally grouped ACEIs and ARBs together, despite growing evidence that their pharmacologic profiles differ. While both agents target the RAS, ARBs would have a more favorable side effect profile, less bradykinin-mediated effects, and potential pleiotropic benefits beyond blood pressure control [17,18]. Yet, no study to date has directly compared the clinical effectiveness of ACEIs and ARBs in patients with MASLD.

To the best of our knowledge, the study is the first to directly compare ACEIs and ARBs in a MASLD-specific population, using a large-scale, real-world dataset with classification by fibrosis and metabolic status. Our findings provide evidences on the differential impact of ACEIs and ARBs in the MASLD population and might have important implications for risk stratification and therapeutic decision-making in clinical practice.

## 2. Results

### 2.1. Baseline Characteristics of the Study Population

The median follow-up duration was 14.8 years (interquartile range [IQR], 13.9–15.6) for the entire cohort. By subgroup, the median follow-up was 14.9 years (IQR, 14.0–15.6) in the No SLD group and 14.8 years (IQR, 13.9–15.6) in the MASLD group. Among 52,143 eligible participants, 15,413 (29.6%) were ARB users and 36,730 (70.4%) were ACEI users. The proportions of ARB and ACEI use were consistent across MASLD and No SLD groups, as well as across age and sex strata (Appendix A). Prior to weighting, notable imbalances were observed in baseline characteristics including age, sex, BMI, liver enzyme levels, and cardiometabolic comorbidities (Appendix A). After inverse IPTW, covariate balance was substantially improved, with SMDs < 0.1 across most variables (Table 1, Appendix A, Appendix A).

### 2.2. Overall Survival in ARB Users Versus ACEI Users in SLD

In the entire cohort, ARB users demonstrated significantly higher overall survival than ACEI users, both before (*p* = 0.003) and after IPTW (*p* = 0.025) (Appendix A). Stratified analysis by liver disease status revealed that this survival advantage was driven predominantly by participants with MASLD. In this classification, ARB use was associated with significantly improved survival both before (*p* = 0.011) and after IPTW (*p* = 0.029) (Figure 1A,B). No significant survival difference was observed between drug classes in the No SLD group (Figure 1C,D).

Multivariable Cox models confirmed these findings. In the entire cohort, ARB use was associated with reduced mortality before (HR, 0.94; 95% CI, 0.90–0.98; *p* = 0.009) and after IPTW (HR, 0.94; 95% CI, 0.90–0.99; *p* = 0.017) (Appendix A). Among participants with MASLD, the mortality benefit of ARB use remained statistically significant (HR, 0.94; 95% CI, 0.90–1.00; *p* = 0.031) after IPTW, whereas no difference was seen in the No SLD group (Table 2, Appendix A).

When examining event rates, MASLD participants using ARBs experienced 1.170 deaths per 100 person-years versus 1.248 in ACEI users, corresponding to an adjusted HR of ACEI users (HR, 1.06; 95% CI, 1.01–1.12; *p* = 0.029). Nevertheless, No SLD participants using ARB showed 0.850 deaths per 100 person-years versus 0.947 in ACEI users, with no significant difference (HR, 1.07; 95% CI, 0.94–1.21; *p* = 0.318) (Appendix A).

### 2.3. Cardiovascular Outcomes and Liver Related Events

ARB use was also associated with a significantly lower risk of cardiovascular events after IPTW in the overall cohort (*p* < 0.001; Appendix A). This benefit was driven by the MASLD group regarding the lower incidence of cardiovascular event of ARB users in the cohort (*p* < 0.001) (Appendix A). No statistically significant difference was observed in the No SLD group (Appendix A).

Cox models further supported these findings. Cardiovascular risk was significantly higher in ACEI users (entire cohort: HR, 1.08; 95% CI, 1.05–1.12; *p* < 0.001; MASLD: HR, 1.09; 95% CI, 1.05–1.13; *p* < 0.001), while the No SLD subgroup showed no significant difference (Table 3). No differences were observed between ARB and ACEI users in terms of hepatic decompensation or HCC incidence across all liver disease cohort (Appendix A).

### 2.4. Subgroup Analysis for Sex, Age, BMI and the Presence of Diabetes Mellitus

Subgroup analyses were conducted by sex, age, BMI, and diabetes status. When stratified by sex, no significant difference in overall survival was observed between ARB and ACEI users in either the MASLD or No SLD groups. Similarly, among age subgroups, ARB use was not associated with improved survival in those under 65 years. However, in participants aged ≥65 years, ARB use was significantly associated with lower mortality in the entire cohort (HR, 0.93; 95% CI, 0.87–1.00; *p* = 0.049), but not within the MASLD or No SLD subgroups. In terms of BMI, ARB use was associated with improved survival in participants with BMI ≥ 25 kg/m^2^. In the entire cohort, the HR was 0.94 (95% CI, 0.89–0.99; *p* = 0.021), and in the MASLD subgroup, a similar benefit was observed (HR, 0.94; 95% CI, 0.90–1.00; *p* = 0.038). No significant mortality difference was observed in those with BMI < 25 kg/m^2^. Regarding diabetes status, ARB use was associated with improved survival in the MASLD subgroup without diabetes (HR, 0.93; 95% CI, 0.87–1.00; *p* = 0.047), whereas no benefit was seen in individuals with diabetes (Appendix A).

For cardiovascular events, ARB use was significantly associated with lower cardiovascular event risk across several subgroups. In male participants, the HRs were 0.88 (95% CI, 0.84–0.92; *p* < 0.001) in the entire cohort and 0.87 (95% CI, 0.84–0.91; *p* < 0.001) in the MASLD subgroup. In participants younger than 65, ARB use was also associated with reduced cardiovascular risk in both the entire cohort (HR, 0.90; 95% CI, 0.86–0.94; *p* < 0.001) and the MASLD subgroup (HR, 0.90; 95% CI, 0.85–0.94; *p* < 0.001). When stratified by BMI, ARB use was associated with lower cardiovascular risk in both groups. For BMI < 25 kg/m^2^, the HRs were 0.89 (95% CI, 0.81–0.97; *p* = 0.013) in the entire cohort and 0.81 (95% CI, 0.69–0.94; *p* = 0.011) in the MASLD subgroup. For BMI ≥ 25 kg/m^2^, the protective effect was even more prominent for statistical significance (entire cohort: HR, 0.93; 95% CI, 0.89–0.96; *p* < 0.001; MASLD: HR, 0.92; 95% CI, 0.89–0.96; *p* < 0.001). Finally, when stratified by diabetes status, ARB use was associated with a significantly lower cardiovascular event risk in participants without diabetes (entire cohort: HR, 0.89; 95% CI, 0.85–0.93; *p* < 0.001; MASLD: HR, 0.88; 95% CI, 0.84–0.92; *p* < 0.001), but no significant difference was observed among individuals with diabetes (Appendix A).

### 2.5. Liver Fibrosis Stratified Outcomes

To assess whether the clinical benefit of ARBs varied by fibrosis severity, stratified analyses were conducted using two noninvasive fibrosis indices, such as FIB-4 and NFS. In the entire cohort, ARB use was significantly associated with improved overall survival in participants without significant fibrosis. Specifically, the adjusted HRs were 0.91 (95% CI, 0.85–0.97; *p* = 0.006) for FIB-4 < 1.45 and 0.89 (95% CI, 0.82–0.97; *p* = 0.006) for NFS < −1.455. No survival advantage was observed among those with significant fibrosis by either index (Appendix A). These findings were presented in Kaplan–Meier curves, with separation favoring ARBs most evident in participants without advanced fibrosis (Appendix A). In the MASLD subgroup, similar trends were observed. ARB use was associated with significantly lower mortality in those with FIB-4 < 1.45 (HR, 0.90; 95% CI, 0.83–0.96; *p* = 0.003) and NFS < −1.455 (HR, 0.88; 95% CI, 0.80–0.97; *p* = 0.011). No significant difference in survival was noted among those with advanced fibrosis (Figure 2A,B, Appendix A).

In contrast, cardiovascular outcomes demonstrated a more consistent and prominent benefit of ARB use across fibrosis severity. In the entire cohort, ARB users had significantly lower cardiovascular event risk in both low and high fibrosis burden groups. For FIB-4 < 1.45, the HR was 0.94 (95% CI, 0.90–0.99; *p* = 0.012), and for FIB-4 ≥ 1.45, it was even stronger (HR, 0.90; 95% CI, 0.86–0.95; *p* < 0.001). Similarly, using the NFS threshold, cardiovascular risk was lower with ARBs for both <–1.455 (HR, 0.93; 95% CI, 0.88–0.98; *p* = 0.009) and ≥−1.455 (HR, 0.92; 95% CI, 0.88–0.96; *p* < 0.001). In the MASLD group, ARB use was consistently protective against cardiovascular events regardless of fibrosis burden. For FIB-4 < 1.45, the HR was 0.93 (95% CI, 0.89–0.98; *p* = 0.004), and for FIB-4 ≥ 1.45, it was 0.90 (95% CI, 0.85–0.95; *p* < 0.001). Using the NFS classification, ARB use conferred significant cardiovascular protection in both lower (HR, 0.91; 95% CI, 0.86–0.97; *p* = 0.004) and higher fibrosis categories (HR, 0.92; 95% CI, 0.88–0.97; *p* = 0.001) (Figure 2C,D, Appendix A). This pattern was also presented with Kaplan–Meier curves (Appendix A). In the No SLD group, neither survival nor cardiovascular outcomes differed significantly between ARB and ACEI users across fibrosis strata (Appendix A).

When advanced fibrosis was defined using rule-in thresholds (FIB-4 ≥ 2.67 or NFS > 0.675), overall prognosis was poorer in patients with advanced fibrosis; however, there was no significant difference between the ARB and ACEI groups (Appendix A).

### 2.6. Differential Spatial Expression of ACEI- and ARB-Target Genes in Mouse Liver

Spatial transcriptomic analysis revealed distinct expression patterns between ACEI- and ARB-target genes. ACEI-target genes (*Ace*, *Kng1*, *Klkb1*, *Bdkrb2*, *Nos3*) demonstrated relatively low overall expression in hepatic tissue, with minimal differences between the Western diet and standard diet groups. In contrast, ARB-target genes (*Agtr1a*, *Agt*, *Tgfbr1*, *Col1a2*, *Pdgfrb*, *Il1b*, *Cybb*) displayed markedly higher expression levels, particularly within fibrotic and pericentral regions of the liver. Notably, several ARB-associated genes were upregulated in Western diet-fed mice compared to standard diet controls, indicating enhanced activation of ARB-related signaling pathways under metabolic stress conditions (Appendix A).

## 3. Discussion

In this population-based cohort study from the UK Biobank, we found that among individuals with MASLD, use of ARBs was associated with significantly improved overall survival and reduced cardiovascular risk compared to ACEIs. This difference persisted after adjustment for confounding using IPTW and was especially pronounced in subgroups including those with elevated body mass index BMI, absence of diabetes, and lower fibrosis burden. Our findings suggest a potential class-specific difference in the effectiveness of RAS inhibitors for MASLD management, especially for ARBs.

While RAS inhibitors have long been investigated for their anti-fibrotic and anti-inflammatory properties, previous studies have grouped ACEIs and ARBs together, despite their differing pharmacologic profiles [8,16]. ACEIs prevent the conversion of angiotensin I to angiotensin II, but they also inhibit the breakdown of bradykinin, potentially contributing to pro-inflammatory effects [19,20]. In contrast, ARBs selectively block the angiotensin II type 1 receptor without affecting bradykinin levels [21]. This distinction might express the differential effects observed in our study, as ARBs are thought to conduct broad antifibrotic, anti-inflammatory, and vasoprotective actions, with potential benefits for both hepatic and cardiovascular outcomes [22,23,24]. A preclinical study had shown that ARBs inhibit hepatic stellate cell activation and collagen deposition more effectively than ACEIs [25].

Our analysis supports these mechanistic insights in a real-world context. The survival benefit of ARBs was more prominent in participants without significant fibrosis (FIB-4 < 1.45 or NFS < −1.455), compared to participants with significant fibrosis, suggesting that early intervention with ARBs might help attenuate disease progression before advanced fibrosis becomes established. Notably, this benefit was not observed in the No SLD group, implying that the observed associations are specific to the presence of steatotic liver disease. In contrast, the cardiovascular protective effect of ARBs was presented across all fibrosis status and was especially prominent in those with advanced fibrosis, highlighting dual action of ARBs in liver and cardiovascular systems.

Subgroup analyses further revealed several insights. Participants without diabetes derived more consistent benefit from ARB use than those with diabetes, both in terms of survival and cardiovascular risk. It is possible that the advanced vascular damage and systemic inflammation often seen in diabetes attenuate the relative benefits of ARB therapy. Meanwhile, ARBs were associated with lower event rates in individuals with higher BMI, a population particularly vulnerable to MASLD progression and cardiovascular events. These findings suggest that ARBs would be especially effective in metabolically active, pre-cirrhotic stages of MASLD.

The transcriptomic data provide molecular support for the observed clinical benefits of ARBs in MASLD. Limited expression of ACEI-target genes suggests a minor role for ACEI-sensitive pathways in disease progression, whereas more prominent induction of ARB-target genes with Western diet conditions presents the pathogenic relevance of angiotensin II receptor signaling. This result is consistent with experimental evidence showing that ARBs exert greater antifibrotic and anti-inflammatory effects than ACEIs and emphasizing ARBs as the more biologically pertinent RAS inhibitors in MASLD [10,25]. Our spatial findings (WD-linked upregulation of *Agtr1a*, *Tgfbr1*, *Col1a2*, *Pdgfrb*, *Il1b*, *Cybb*) are consistent with a body of evidence that ARBs attenuate angiotensin II type 1 receptor (AGTR1), transforming growth factor β (TGF-β)/platelet-derived growth factor (PDGF) profibrotic signaling and downstream collagen deposition (Collagen 1A1/2) in steatohepatitis and cholestatic fibrosis models [26,27,28]. ARBs also reduce IL-1β expression and macrophage inflammatory programs and dampen oxidative stress via NADPH oxidase pathways [29,30]. In contrast, ACEIs primarily act upstream by limiting angiotensin II generation while enhancing bradykinin/NO tone [31,32]. These might offer vasoprotective effects but appear less effective at suppressing stellate cell proliferation and matrix gene induction in fatty-liver models relative to ARBs. Collectively, these data help explain why ARBs showed stronger associations with survival and cardiovascular outcomes in MASLD in our study.

Nonetheless, several limitations might be acknowledged. Medication exposure was based on self-report at baseline and would not capture treatment adherence or longitudinal changes in therapy. Additionally, unmeasured confounding remains a possibility as an observational study. ACEIs may be more commonly prescribed to patients with higher cardiovascular risk, such as those with heart failure or post-myocardial infarction, potentially biasing results in favor of ARBs. Although our IPTW models adjusted for key cardiometabolic covariates, residual confounding might persist. Moreover, we cannot exclude the possibility of differential effects within subclasses of ARBs or ACEIs, which were grouped by class for statistical power. In addition, medication exposure was self-reported at baseline, and information on dose, adherence, or treatment changes over time was unavailable. This limitation precluded a new-user or time-varying design and might contribute to residual confounding or immortal-time bias. And the mouse Liver Cell Atlas analysis was exploratory and not directly linked to the causal inference drawn from human data. These findings should therefore be interpreted as hypothesis-generating and supportive rather than confirmatory. Lastly, fibrosis severity was assessed using surrogate biomarkers rather than histology, although both FIB-4 and NFS are well validated and commonly used in population studies.

The spatial transcriptomic data analyzed here were derived from mouse liver sections, and cross-species extrapolation warrants caution. Nevertheless, the observed pattern, low diet-responsiveness of ACEI-related targets and strong WD-associated induction of ARB-related fibrotic, inflammatory readouts, addresses conserved non-parenchymal compartments reported in human liver atlases, including macrophage and stellate-cell niches that harbor AGTR1- and PDGF receptor beta-linked programs during chronic injury. Importantly, multiple preclinical and translational studies in steatohepatitis and fibrosis models show that AGTR1 blockade reduces TGF-β signaling, collagen transcripts (Collagen 1A1/2), PDGF receptor beta-dependent stellate-cell activation, IL-1β-driven inflammation, and nicotinamide adenine dinucleotide phosphate oxidase activity supporting biological plausibility in humans [33]. Future work would directly validate these gene-level signatures in human MASLD tissues and test whether class-specific modulation persists after adjustment for indication and dose.

The present findings have potential clinical implications. As MASLD continues to rise in prevalence worldwide, identifying therapies that target both liver-related and systemic outcomes is increasingly important. Our results suggest that ARBs would represent a more favorable RAS-inhibitor class for patients with MASLD, particularly in those with early-stage disease and elevated metabolic risk.

## 4. Materials and Methods

### 4.1. Study Design and Population

The present study was conducted using data from the UK Biobank (Application ID: 117214, Approval date: 25 September 2023), which includes over 500,000 participants aged 40–69 years at baseline (2006–2010), with linked health, laboratory, and genetic data. Ethical approval was granted by the North West Multi-Centre Research Ethics Committee (REC reference 11/NW/0382, Approval date: 31 May 2011) [34].

From the original 502,370 participants, individuals with chronic liver diseases, including autoimmune hepatitis, chronic hepatitis B or C, primary biliary cholangitis, and cryptogenic steatotic liver disease, were excluded. Participants with alcohol-associated liver disease and metabolic dysfunction and alcohol-associated liver disease were also removed to isolate a metabolically driven population. Cryptogenic steatotic liver disease was defined as fatty liver in the absence of viral, alcohol-related, or autoimmune etiologies. After excluding individuals who used both or neither of ACEIs and ARBs, the final analytic cohort included 52,143 participants: 15,413 ARB users and 36,730 ACEI users (Figure 3).

Participants were categorized as having MASLD or no steatotic liver disease (No SLD) based on the fatty liver index (FLI). FLI was calculated using the following validated formula [35]:FLI = (e ^0.953 × loge (triglycerides) + 0.139 × BMI + 0.718 × loge [gamma-glutamyl transpeptidase (ggt)] + 0.053 × waist circumference − 15.745^)/(1 + e ^0.953 × loge (triglycerides) + 0.139 × BMI + 0.718 × loge (ggt) + 0.053 × waist circumference − 15.745^) × 100

An FLI cutoff of ≥60 was used to define hepatic steatosis [35].

MASLD was defined based on the recent multisociety Delphi consensus statement as the presence of hepatic steatosis, determined by FLI ≥ 60 in this study, in combination with at least one cardiometabolic risk factor. These included overweight or obesity (BMI ≥ 25 kg/m^2^), type 2 diabetes mellitus, pre-diabetes (impaired fasting glucose or impaired glucose tolerance), hypertension (or use of antihypertensive medication), and dyslipidemia (defined as hypertriglyceridemia, low high-density lipoprotein-cholesterol, or use of lipid-lowering agents) [36].

Cardiovascular events were identified using UK Biobank Field IDs 131296, 131298, 131300, 131302, 131304, and 131306 from Hospital Episode Statistics and death records. Hepatic decompensation included varices with or without bleeding (I85.0, I85.9, I98.2, I98.3), portal hypertension (K76.6), hepatorenal syndrome (K76.7), and ascites (R18). Hepatocellular carcinoma (HCC) was defined by ICD-10 code C22.0, and date of death was obtained from Field ID 40000. Outcomes were identified algorithmically through linkage with national hospital and death registries, without individual adjudication.

### 4.2. Noninvasive Assessment of Liver Fibrosis

Liver fibrosis was assessed using two validated noninvasive scoring systems: Fibrosis-4 (FIB-4) index and the NAFLD fibrosis score (NFS). These indices have been widely validated against liver biopsy and other fibrosis measures.

FIB-4 was calculated as follows [37]:FIB-4 = age (years) × AST [U/L]/(platelets [10^9^/L] × (ALT [U/L])^1/2^) 

This score was initially developed and validated as a simple and accurate marker of liver fibrosis in patients with hepatitis C virus infection, showing good correlation with liver biopsy and the FibroTest^®^ [37].

NFS was calculated as follows [38]:NFS = −1.675 + 0.037 × age (years) + 0.094 × BMI (kg/m^2^) + 1.13 × IFG/diabetes (yes = 1, no = 0) + 0.99 × AST/ALT ratio − 0.013 × platelet (×10^9^/L) − 0.66 × albumin (g/dL) 

The NFS was derived from a large cohort of patients with biopsy-proven NAFLD and has been validated to reliably exclude advanced fibrosis in clinical settings [38]. Significant fibrosis was defined as FIB-4 ≥ 1.45 or NFS ≥ −1.455, thresholds that provide high negative predictive value for advanced fibrosis and are recommended for identifying low-risk patients [37,38].

### 4.3. Medication Exposure Definition

Medication use was determined using self-reported data from UK Biobank Data-Field 20003, which records the names and codes of medications taken regularly at baseline. Only participants who reported exclusive use of either an ACEI or an ARB were included. Those using both classes simultaneously, or using neither, were excluded to allow for a clean comparison of the two drug classes.

ARBs included a broad range of agents such as losartan (1140916356), valsartan (1141145660), irbesartan (1141152998), candesartan cilexetil (1141156836), telmisartan (1141166006), eprosartan (1141171336), and olmesartan (1141193282). Similarly, ACEIs included commonly prescribed agents such as lisinopril (1140860696), enalapril (1140888552), ramipril (1140860806), captopril (1140860750), quinapril (1140860728), and perindopril (1140888560).

All drug identifications were made based on the UK Biobank’s medication coding system (Appendix A).

### 4.4. Spatial Transcriptomic Data from the Liver Cell Atlas

We queried spatial transcriptomic datasets from the Liver Cell Atlas (https://www.livercellatlas.org) derived from mouse livers under standard diet (SD, visualized in purple) and Western diet (WD, visualized in yellow) conditions. Pre-specified gene sets were assembled a priori to represent pharmacological targets or pathway readouts of ACE inhibitors (*Ace*, *Kng1*, *Klkb1*, *Bdkrb2*, *Nos3*) and ARBs (*Agtr1a*, *Agt*, *Tgfbr1*, *Col1a2*, *Pdgfrb*, *Il1b*, *Cybb*) based on prior mechanistic literature implicating the RAS, bradykinin/nitric oxide (NO) signaling, and profibrotic, inflammatory axes in steatotic liver disease [26,27,28,29,30,31,32].

Analyses were performed in Python 3.10 using Scanpy 1.9.x and Squidpy 1.2.x. Raw count matrices were library-size normalized (counts per 10,000), log1p-transformed, and inspected for spatial quality control (spot count depth, mitochondrial fraction). Per-spot gene set scores (ACEI-target set; ARB-target set) were computed as the average z-scored expression of member genes after subtracting a matched control gene set (Scanpy tl.score_genes). Group comparisons (WD versus SD) used spot-level linear models with section ID as a random effect (statsmodels mixed effects), and Benjamini–Hochberg false discovery rate control (q < 0.05). Spatial localization was summarized by per-region score aggregation (periportal, mid-lobular, pericentral masks provided with the atlas. If it is unavailable, pericentral zones were approximated by high *Cyp2e1*/*Glul* expression). Differential expression for individual genes used rank-based tests (Wilcoxon) with FDR control. Figure was rendered with Matplotlib; color keys explicitly map SD to purple and WD to yellow for clarity (Appendix A).

### 4.5. Statistical Analysis

Baseline characteristics were compared between ARB and ACEI users before and after inverse probability of treatment weighting (IPTW), using standardized mean differences (SMDs). Propensity scores were estimated using logistic regression, including variables such as age, sex, body mass index (BMI), physical activity, smoking, alcohol intake, diabetes, hypertension, dyslipidemia, blood pressure, liver function tests, platelet count, and albumin. The 10-year ASCVD risk score was calculated based on pooled cohort equations and summarized as baseline characteristics.

After applying IPTW, Kaplan–Meier survival curves and log-rank tests were used to assess overall survival and cardiovascular event-free survival. Cox proportional hazards models were used to calculate hazard ratios (HRs) and 95% confidence intervals (CIs) for overall mortality, cardiovascular disease, hepatic decompensation, and HCC. Multivariable models adjusted for covariates with residual imbalance or clinical relevance.

Subgroup analyses were performed by sex, age group (<65 vs. ≥65), BMI category (<25 vs. ≥25 kg/m^2^), and diabetes status. Classified analyses were also conducted based on fibrosis severity (using both FIB-4 and NFS) to assess effect modification. All statistical analyses were performed using R version 4.2.1 (R Foundation for Statistical Computing, Vienna, Austria), with two-tailed *p* values < 0.05 considered statistically significant.

## 5. Conclusions

In conclusion, our study demonstrates that ARB use is associated with improved survival and reduced cardiovascular events compared to ACEI use in patients with MASLD. These findings highlight important differences in the real-world effectiveness of RAS inhibitor subclasses and emphasize the need for personalized treatment strategies.

## Figures and Tables

**Figure 1 ijms-26-10061-f001:**
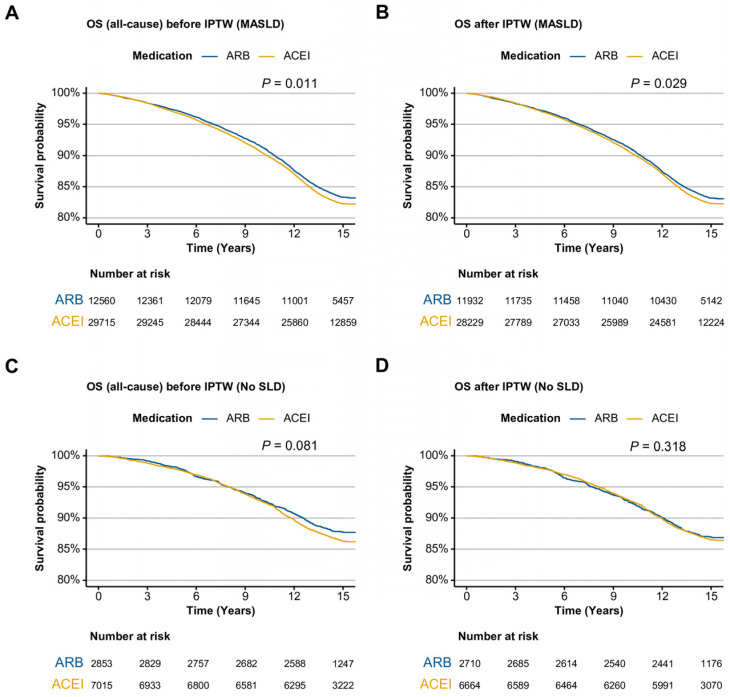
Kaplan–Meier curves of ARB and ACEI users for all-cause mortality in the MASLD and No SLD cohorts. (**A**) MASLD cohort before IPTW. (**B**) MASLD cohort after IPTW. (**C**) No SLD cohort before IPTW. (**D**) No SLD cohort after IPTW. ARB, angiotensin II receptor blocker; ACEI, angiotensin-converting enzyme inhibitor; OS, overall survival, IPTW, inverse probability of treatment weighting; SLD, steatotic liver disease; MASLD, metabolic dysfunction-associated steatotic liver disease.

**Figure 2 ijms-26-10061-f002:**
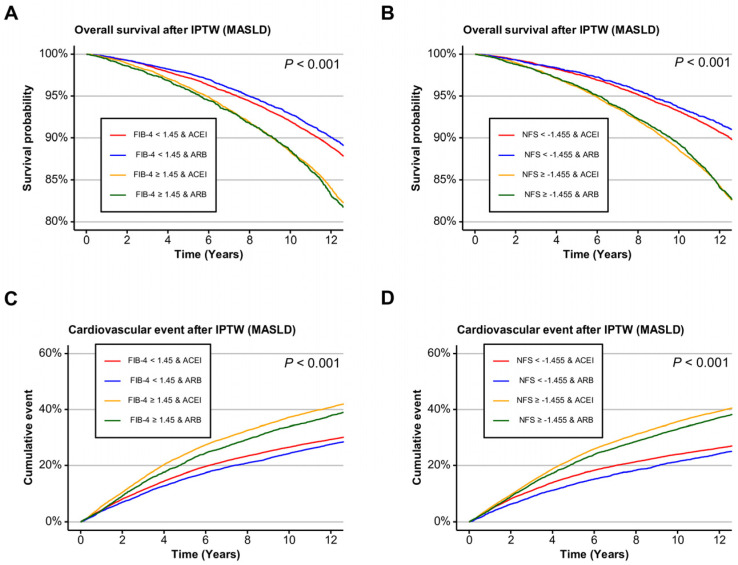
Kaplan–Meier curves for all-cause mortality and incidence of cardiovascular events of ARB and ACEI users, with or without significant liver fibrosis in the MASLD cohort after IPTW. (**A**) all-cause mortality with FIB-4 < 1.45 or ≥1.45. (**B**) all-cause mortality with NFS < −1.455 or ≥−1.455. (**C**) Cardiovascular event incidence users with FIB-4 < 1.45 or ≥1.45. (**D**) Cardiovascular event incidence with NFS < −1.455 or ≥−1.455. ARB, angiotensin II receptor blocker; ACEI, angiotensin-converting enzyme inhibitor; IPTW, inverse probability of treatment weighting; FIB-4, fibrosis-4; NFS, non-alcoholic fatty liver disease fibrosis score; MASLD, metabolic dysfunction-associated steatotic liver disease.

**Figure 3 ijms-26-10061-f003:**
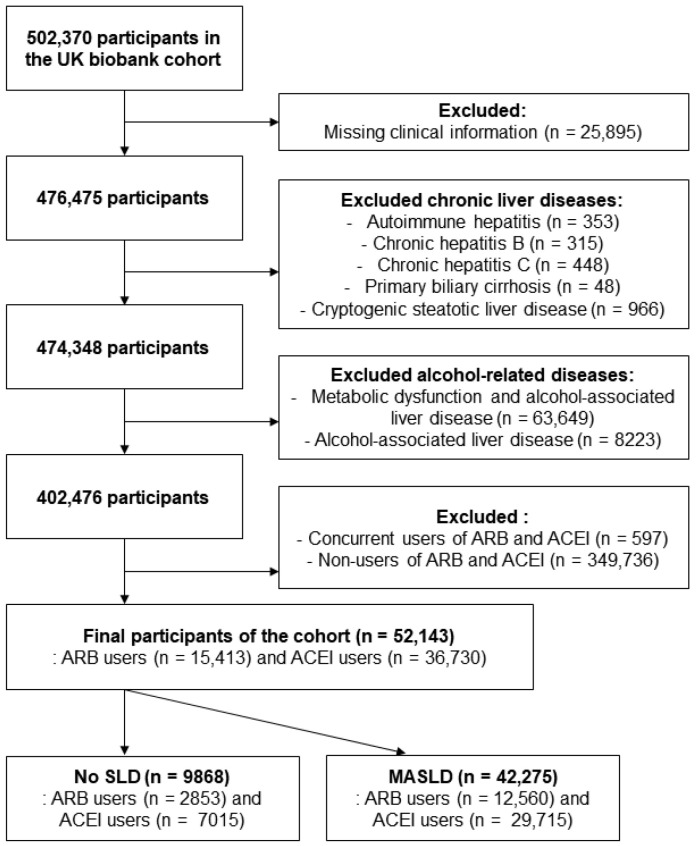
Flow chart of population stratification. From the UK Biobank cohort, participants with missing clinical data or with a history of autoimmune hepatitis, chronic viral hepatitis, primary biliary cholangitis, cryptogenic steatotic liver disease, metabolic dysfunction and alcohol-associated liver disease, or alcohol-associated liver disease alone were excluded. The remaining individuals were stratified into two groups based on steatosis status: No SLD and MASLD. Each group was further categorized into ARB and ACEI user subgroups. ARB, angiotensin II receptor blocker; ACEI, angiotensin-converting enzyme inhibitor; SLD, steatotic liver disease; MASLD, metabolic dysfunction-associated steatotic liver disease.

**Table 1 ijms-26-10061-t001:** Baseline clinical characteristics after IPTW.

Clinical Characteristics	No SLD			MASLD			
	Medication		SMD	Medication		SMD	
	ARB, N = 2710	ACEI, N = 6664		ARB, N = 11,932	ACEI, N = 28,229		
**Sex (Male)**	863 (31.8%)	2123 (31.9%)	0.001	7053 (59.1%)	16,713 (59.2%)	0.002	
**Age at recruitment**	60.70 ± 6.61	60.71 ± 6.58	0.003	60.38 ± 6.59	60.38 ± 6.61	<0.001	
**Physical activity**			0.002			0.004	
More than 4 times	2290 (84.5%)	5624 (84.4%)		9066 (76.0%)	21,490 (76.1%)		
Under 4 times	421 (15.5%)	1040 (15.6%)		2866 (24.0%)	6739 (23.9%)		
**Body mass index (kg/m^2^)**	23.92 ± 2.31	23.94 ± 2.25	0.008	31.39 ± 5.00	31.36 ± 5.24	0.005	
**Waist circumference (cm)**	79.78 ± 7.26	79.82 ± 7.08	0.006	101.86 ± 12.09	101.82 ± 12.31	0.003	
**Type 2 diabetes**	320 (11.8%)	794 (11.9%)	0.003	4195 (35.2%)	9906 (35.1%)	0.001	
**Dyslipidemia**	1476 (54.4%)	3629 (54.4%)	<0.001	8289 (69.5%)	19,607 (69.5%)	<0.001	
**Hypertension**	2622 (96.7%)	6449 (96.8%)	0.002	11,672 (97.8%)	27,598 (97.8%)	0.004	
**ALT (U/L)**	18.82 ± 7.47	18.87 ± 7.78	0.006	27.53 ± 14.86	27.52 ± 16.72	<0.001	
**GGT (U/L)**	24.15 ± 15.10	24.17 ± 14.94	0.001	46.90 ± 46.72	46.89 ± 45.93	<0.001	
**Platelet (10^9^/L)**	253.35 ± 62.08	253.11 ± 63.31	0.004	251.57 ± 62.71	251.59 ± 62.62	<0.001	
**Albumin (g/L)**	4.53 ± 0.27	4.53 ± 0.26	0.001	4.51 ± 0.26	4.51 ± 0.26	0.002	
**FIB-4 score ≥ 1.45**	1311 (48.4%)	3222 (48.4%)	<0.001	4545 (38.1%)	10,727 (38.0%)	0.002	
**NFS ≥ −1.455**	884 (32.6%)	2202 (33.0%)	0.009	6752 (56.6%)	15,918 (56.4%)	0.004	
**ASCVD risk score**	6.88 ± 6.11	6.95 ± 6.18	0.007	16.24 ± 13.44	16.03 ± 13.28	0.015	

Data are described as mean ± standard deviation or N (%). IPTW, inverse probability of treatment weighting; SLD, steatotic liver disease; MASLD, metabolic dysfunction-associated steatotic liver disease; ARB, angiotensin receptor blocker; ACEI, angiotensin-converting enzyme inhibitor; ALT, alanine transaminase; GGT, gamma-glutamyl transferase; SMD, standardized mean difference; FIB-4, fibrosis-4; NFS, NAFLD fibrosis score; ASCVD, atherosclerotic cardiovascular disease.

**Table 2 ijms-26-10061-t002:** Univariate and multivariate analyses of overall survival before and after IPTW in MASLD patients.

Before IPTW						
Clinical Characteristics	Univariate Analysis			Multivariate Analysis		
	HR	95% CI	*p* Value	HR	95% CI	*p* Value
**Sex (Male)**	1.54	1.46, 1.62	<0.001	1.38	1.29, 1.47	<0.001
**Age at recruitment**	1.09	1.08, 1.09	<0.001	1.08	1.08, 1.09	<0.001
**Physical activity (Under 4 times)**	1.21	1.15, 1.28	<0.001	1.18	1.12, 1.24	<0.001
**Body mass index (kg/m^2^)**	1.02	1.01, 1.02	<0.001	0.98	0.97, 0.99	<0.001
**Waist circumference (cm)**	1.02	1.02, 1.02	<0.001	1.02	1.02, 1.02	<0.001
**Type 2 diabetes**	1.75	1.67, 1.83	<0.001	1.51	1.43, 1.59	<0.001
**Dyslipidemia**	1.53	1.45, 1.62	<0.001	1.08	1.02, 1.14	0.013
**ALT (U/L)**	0.99	0.99, 1.0	<0.001	0.99	0.99, 0.99	<0.001
**GGT (U/L)**	1.00	1.00, 1.00	<0.001	1.00	1.00, 1.00	<0.001
**Platelet (10^9^/L)**	1.00	1.00, 1.00	<0.001	1.00	1.00, 1.00	<0.001
**Albumin (g/L)**	0.40	0.37, 0.43	<0.001	0.51	0.47, 0.56	<0.001
**FIB-4 score ≥ 1.45**	1.56	1.49, 1.63	<0.001	1.16	1.09, 1.23	<0.001
**NFS ≥ −1.455**	1.88	1.79, 1.98	<0.001	1.03	0.96, 1.11	0.421
**ASCVD risk score**	1.03	1.03, 1.03	<0.001	1.01	1.00, 1.01	<0.001
**ARB user**	0.94	0.89, 0.98	0.011	0.94	0.89, 0.99	0.023
**After IPTW**						
**Clinical Characteristics**	**Univariate Analysis**			**Multivariate Analysis**		
	**HR**	**95% CI**	***p* Value**	**HR**	**95% CI**	***p* Value**
**Sex (Male)**	1.53	1.46, 1.61	<0.001	1.39	1.30, 1.48	<0.001
**Age at recruitment**	1.09	1.08, 1.10	<0.001	1.08	1.08, 1.09	<0.001
**Physical activity (Under 4 times)**	1.21	1.15, 1.28	<0.001	1.18	1.12, 1.24	<0.001
**Body mass index (kg/m^2^)**	1.02	1.01, 1.02	<0.001	0.98	0.97, 0.99	<0.001
**Waist circumference (cm)**	1.02	1.02, 1.02	<0.001	1.02	1.02, 1.02	<0.001
**Type 2 diabetes**	1.75	1.67, 1.83	<0.001	1.50	1.42, 1.59	<0.001
**Dyslipidemia**	1.54	1.45, 1.62	<0.001	1.08	1.02, 1.14	0.009
**ALT (U/L)**	0.99	0.99, 1.00	<0.001	0.99	0.99, 0.99	<0.001
**GGT (U/L)**	1.00	1.00, 1.00	<0.001	1.00	1.00, 1.00	<0.001
**Platelet (10^9^/L)**	1.00	1.00, 1.00	<0.001	1.00	1.00, 1.00	<0.001
**Albumin (g/L)**	0.4	0.36, 0.43	<0.001	0.51	0.46, 0.56	<0.001
**FIB-4 score ≥ 1.45**	1.55	1.48, 1.63	<0.001	1.16	1.09, 1.24	<0.001
**NFS ≥ −1.455**	1.88	1.78, 1.97	<0.001	1.03	0.96, 1.12	0.406
**ASCVD risk score**	1.03	1.03, 1.03	<0.001	1.01	1.00, 1.01	<0.001
**ARB user**	0.95	0.90, 1.00	0.029	0.94	0.90, 1.0	0.031

IPTW, inverse probability of treatment weighting; HR, hazard ratio; CI, confidence interval; ALT, alanine aminotransferase (upper normal limit = 40 U/L); GGT, gamma-glutamyl transferase (upper normal limit = 50 U/L); FIB-4, fibrosis-4; NFS, NAFLD fibrosis score; ASCVD, atherosclerotic cardiovascular disease; ARB, angiotensin receptor blocker.

**Table 3 ijms-26-10061-t003:** IPTW-adjusted comparison of cardiovascular events between ARB and ACEI users.

Medication	Entire Cohort				
	Number of Patients	Number of Events	Number of Events/ 100 Patient-Years	HR (95% CI)	*p* Value
**ARB**	15,413	4665	2.584	Reference	
**ACEI**	36,730	12,682	3.088	1.08 (1.05–1.12)	<0.001
**Medication**	**No SLD**				
	**Number of Patients**	**Number of Events**	**Number of Events/** **100 Patient-Years**	**HR (95% CI)**	***p* Value**
**ARB**	2853	639	1.802	Reference	
**ACEI**	7015	1837	2.199	1.07 (0.97–1.17)	0.162
**Medication**	**MASLD**				
	**Number of Patients**	**Number of Events**	**Number of Events/** **100 Patient-Years**	**HR (95% CI)**	***p* Value**
**ARB**	12,560	4026	2.775	Reference	
**ACEI**	29,715	10,845	3.315	1.09 (1.05–1.13)	<0.001

IPTW, inverse probability of treatment weighting; ARB, angiotensin receptor blocker; ACEI, angiotensin-converting enzyme inhibitor; HR, hazard ratio; CI, confidence interval; SLD, steatotic liver disease; MASLD, metabolic dysfunction-associated steatotic liver disease.

## Data Availability

The data underlying this study are available from the UK Biobank (https://www.ukbiobank.ac.uk/) upon application and approval.

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
