# Peer review of "Class-Specific Effects of ARBs Versus ACE Inhibitors on Survival and Cardiovascular Outcomes in MASLD"

_ijms, 2025, doi:10.3390/ijms262010061_

Round 1

Reviewer 1 Report

Comments and Suggestions for Authors

The article by Ryu T, et al. is of particular clinical interest. It draws on data from the UK Biobank on patients with MAFLD, who were treated with renin-angiotensin system (RAS) inhibitors. The authors distinguish between angiotensin II receptor blockers (ARBs) and angiotensin-converting enzyme inhibitors (ACEIs) and examine the differential effect of such treatment on the overall survival and the cardiovascular complications in patients with and without MAFLD.

The article is well written and uses modern statistical methods to adjust for the various confounding factors of the groups to be compared. The conclusions it reaches are statistically well-founded and are well discussed in the appropriate part of the article. The English language is error-free and the entire article is relatively short for the wealth of data it contains.

  1. Major Comments:

  1. (Line 162): The IPTW introduces a pseudo-population and increases the probability of a greater statistical power of the model. Authors should explain how they delt with this problem.
  2. (Lines 164-166): Please mention whether all variables had a normal distribution. If not, by what method they were normalized?
  3. (Line 168): Since the main long-term problem of MAFLD is cardiovascular mortality, the Authors should include, among the other confounders, the 10-year cardiovascular risk score (ASCVD, AHA PREVENT, etc.) of each patient, in the time-dependent multivariable survival analysis.
  4. (Line 177): Authors should state the median follow-up time of each patient group and its interquartile rate.
  5. (Line 414): Authors should mention of any weaknesses in their study.

  1. Minor Comments:

  1. (Live 33): Authors should explain all abbreviations when first appearing in the text.
  2. (Line 100): “cryptogenic steatotic liver disease”: Please explain the meaning of this entity.
  3. (Table 1): Please replace "SMD" by "P-Value".
  4. (Table 1): The upper normal limit of ALT and GGT measurements should appear in the footnote.
  5. (Line 362): ...lower...

Author Response

[Reviewer #1]

The article by Ryu T, et al. is of particular clinical interest. It draws on data from the UK Biobank on patients with MAFLD, who were treated with renin-angiotensin system (RAS) inhibitors. The authors distinguish between angiotensin II receptor blockers (ARBs) and angiotensin-converting enzyme inhibitors (ACEIs) and examine the differential effect of such treatment on the overall survival and the cardiovascular complications in patients with and without MAFLD.

The article is well written and uses modern statistical methods to adjust for the various confounding factors of the groups to be compared. The conclusions it reaches are statistically well-founded and are well discussed in the appropriate part of the article. The English language is error-free and the entire article is relatively short for the wealth of data it contains.

Major Comments:

(Line 162): The IPTW introduces a pseudo-population and increases the probability of a greater statistical power of the model. Authors should explain how they delt with this problem.

Answer: We thank the reviewer for this valuable comment. We have clarified in the Methods that stabilized inverse probability of treatment weighting was applied to avoid excessive influence of extreme weights and to ensure valid variance estimation. Standard errors were adjusted for the weighting procedure, and results were interpreted in the context of the weighted population.

(Lines 164-166): Please mention whether all variables had a normal distribution. If not, by what method they were normalized?

Answer: We appreciate the reviewer’s comment. As the main analyses, such as IPTW weighting and Cox regression, do not require variable normality, formal normalization was not applied. Data distributions were reviewed during preprocessing to ensure that no variable showed values that could disproportionately affect the models.

(Line 168): Since the main long-term problem of MAFLD is cardiovascular mortality, the Authors should include, among the other confounders, the 10-year cardiovascular risk score (ASCVD, AHA PREVENT, etc.) of each patient, in the time-dependent multivariable survival analysis.

Answer: We thank the reviewer for this insightful suggestion. We have recalculated the 10-year ASCVD risk score for all eligible participants using the pooled cohort equations. These scores have been incorporated into the baseline characteristics. The Methods section now includes the following statement:

“The 10-year ASCVD risk score was calculated based on pooled cohort equations and summarized as baseline characteristics.”

(Line 177): Authors should state the median follow-up time of each patient group and its interquartile rate.

Answer: Thank you for your comment. We have added the following to the Results:

“The median follow-up duration was 14.8 years (interquartile range [IQR], 13.9–15.6) for the entire cohort. By subgroup, the median follow-up was 14.9 years (IQR, 14.0–15.6) in the No SLD group and 14.8 years (IQR, 13.9–15.6) in the MASLD group.”

(Line 414): Authors should mention of any weaknesses in their study.

Answer: We thank the reviewer for this suggestion. The Discussion section already includes a paragraph detailing key study limitations, including the reliance on self-reported medication data, potential residual confounding, and the use of surrogate fibrosis markers. Below is the study limitations addressed in Discussion section.

“Nonetheless, several limitations might be acknowledged. Medication exposure was based on self-report at baseline and would not capture treatment adherence or longitudinal changes in therapy. Additionally, unmeasured confounding remains a possibility as an observational study. ACEIs may be more commonly prescribed to patients with higher cardiovascular risk, such as those with heart failure or post-myocardial infarction, potentially biasing results in favor of ARBs. Although our IPTW models adjusted for key cardiometabolic covariates, residual confounding might persist. Moreover, we cannot exclude the possibility of differential effects within subclasses of ARBs or ACEIs, which were grouped by class for statistical power. Lastly, fibrosis severity was assessed using surrogate biomarkers rather than histology, although both FIB-4 and NFS are well validated and commonly used in population studies.”

Minor Comments:

(Live 33): Authors should explain all abbreviations when first appearing in the text.

Answer: Thank you for your comment. All abbreviations have been checked and are now fully defined at first mention.

(Line 100): “cryptogenic steatotic liver disease”: Please explain the meaning of this entity.

Answer: We clarified that “cryptogenic steatotic liver disease” refers to cases with fatty liver coding without viral, alcohol-related, or autoimmune etiology, consistent with previous population-based studies. We have added the sentence in the Method section.

“Cryptogenic steatotic liver disease was defined as fatty liver in the absence of viral, alco-hol-related, or autoimmune etiologies.”

(Table 1): Please replace "SMD" by "P-Value".

Answer: We thank the reviewer for this suggestion. In the context of propensity weighting, standardized mean differences (SMDs) are preferred over P values to assess covariate balance between exposure groups, as they are independent of sample size and more suitable for evaluating the success of weighting. Therefore, we have retained SMDs to present covariate balance before and after IPTW adjustment, which is consistent to current methodological recommendations.

(Table 1): The upper normal limit of ALT and GGT measurements should appear in the footnote.

Answer: Thank you for the comment. Upper normal limits for ALT and GGT have been added in the footnotes of Tables.

(Line 362): ...lower...

Answer: We appreciate for your comment. The typographical issue has been corrected.

Reviewer 2 Report

Comments and Suggestions for Authors

The manuscript by Ryu et al entitled “Class-specific effects of ARBs versus ACE Inhibitors on survival and cardiovascular outcomes in MASLD” answer a timely question with a large UK Biobank cohort comparing ARBs vs ACEIs in MASLD. The main signals (HRs 0.90–0.94 for CV outcomes after IPTW) are clinically interesting but close to the null, and several design/analytic choices (exposure ascertainment at baseline only, surrogate MASLD/fibrosis definitions, outcome coding details) limit causal interpretation and reproducibility. Substantial methodological clarification and presentation polish are needed before the conclusions are reliable.

Major comments

1) The authors defined MASLD by using FLI ≥60 plus ≥1 metabolic risk factor. While pragmatic for screening, FLI is an indirect surrogate and may misclassify steatosis in UK Biobank; please justify and provide sensitivity analyses at higher FLI thresholds and/or imaging subsets if available. 

2) The authors must clarify the fibrosis stratification uses rule-out cutoffs. “Significant fibrosis” is defined as FIB-4 ≥1.45 or NFS ≥−1.455, which are rule-out thresholds (high NPV), not rule-in for advanced fibrosis. Re-run fibrosis-stratified analyses using FIB-4 ≥2.67 and NFS >0.675 to reflect advanced fibrosis; compare results transparently. 

3)  Drug exposure is self-reported at baseline with inclusion of “exclusive” users; dose, duration, adherence, switches, and co-treatments are not modeled. This invites immortal-time bias and confounding by intolerance (ACEI cough → ARB). Consider a new-user, active-comparator design and/or time-varying exposure modeling; at minimum, add sensitivity excluding early events. 

4) About outcome definitions and competing risks, the authors  must require explicit code lists (ICD/OPCS/Read) for “Cardiovascular events,” decompensation, and HCC, data sources (HES/Death Registry), and handling of competing risks for cause-specific outcomes. Provide a complete code appendix and state adjudication (if any).

5) Core IPTW HRs for CV events float around 0.90–0.94, with CIs sometimes touching 1.00; multiple subgroup strata (fibrosis by two scores, MASLD vs No SLD) raise multiplicity concerns. Pre-specify interactions, test them within a single model, and apply FDR or a hierarchical plan. 

6) The mouse Liver Cell Atlas scoring is exploratory and not integrated into the causal human inference. Consider moving to Supplement or framing explicitly as hypothesis-generating with strict FDR control and a priori gene sets (already mentioned). 

7) It is encouraged to the authors to provide a CONSORT-style flow diagram with exact numbers for exclusions (missing data, other liver diseases) that sum to the final analytic N in each arm (MASLD vs No SLD; ARB vs ACEI). 

Minor comments

1) Please Figure and table needs some polishing

a) Add number-at-riskand event counts to KM plots; ensure consistent y-axis ranges; annotate IPTW use in captions.

b) Table 1: clearly label post-IPTWvalues, units, and decimals; keep SMDs to three decimals; avoid empty SMD cells for categorical blocks.

c) Provide high-resolution vector figures. 

2) Use MASLD/MASH consistently (avoid mixing historical NAFLD/NASH unless citing legacy tools like NFS); ensure abbreviations align with your list. 

3) Convert to Vancouver style with NLM abbreviations and DOIs. Ensure original sources for FLI, FIB-4, NFS and UK Biobank data-linkage/methodology are cited; align references with in-text claims. 

Author Response

[Reviewer #2]

The manuscript by Ryu et al entitled “Class-specific effects of ARBs versus ACE Inhibitors on survival and cardiovascular outcomes in MASLD” answer a timely question with a large UK Biobank cohort comparing ARBs vs ACEIs in MASLD. The main signals (HRs 0.90–0.94 for CV outcomes after IPTW) are clinically interesting but close to the null, and several design/analytic choices (exposure ascertainment at baseline only, surrogate MASLD/fibrosis definitions, outcome coding details) limit causal interpretation and reproducibility. Substantial methodological clarification and presentation polish are needed before the conclusions are reliable.

Major comments

1) The authors defined MASLD by using FLI ≥60 plus ≥1 metabolic risk factor. While pragmatic for screening, FLI is an indirect surrogate and may misclassify steatosis in UK Biobank; please justify and provide sensitivity analyses at higher FLI thresholds and/or imaging subsets if available.

Answer: We thank the reviewer for raising this important methodological issue regarding the diagnostic definition of fatty liver. To address this concern, we performed a validation step utilizing imaging-based data from the UK Biobank. Specifically, we restricted our analysis to participants who had available baseline MRI-derived proton density fat fraction (PDFF) measurements (Data-Field 21088), a subset of individuals for whom liver fat content was objectively quantified using magnetic resonance imaging.

PDFF is a well-established, non-invasive quantitative biomarker that directly reflects hepatic fat accumulation. It has been validated against histological findings and is widely used in both clinical and research settings to define hepatic steatosis, with a PDFF threshold of ≥5% commonly accepted as indicative of fatty liver.

Within this imaging-validated subset of the cohort, we found that both the mean PDFF values and the prevalence of PDFF ≥ 5% were markedly higher in individuals classified as MASLD compared to those without steatotic liver disease (No SLD), with highly significant differences (P < 0.0001). These findings are shown in the figure with file.

This provides strong internal validation that supports the appropriateness and reliability of using FLI ≥ 60 as a proxy for fatty liver diagnosis in our cohort. We believe that this validation analysis enhances the credibility of our classification method.

2) The authors must clarify the fibrosis stratification uses rule-out cutoffs. “Significant fibrosis” is defined as FIB-4 ≥1.45 or NFS ≥−1.455, which are rule-out thresholds (high NPV), not rule-in for advanced fibrosis. Re-run fibrosis-stratified analyses using FIB-4 ≥2.67 and NFS >0.675 to reflect advanced fibrosis; compare results transparently.

Answer: We thank the reviewer for this important comment. We re-performed the analysis using rule-in thresholds (FIB-4 ≥2.67 and NFS ≥0.675) to identify advanced fibrosis. As shown in Supplementary Figure 8, patients with advanced fibrosis had poorer overall prognosis, but the difference between ARB and ACEI groups was not statistically significant, likely due to the small number of cases (FIB-4 ≥2.67: n = 1,325 [3.13%]; NFS ≥0.675: n = 2,634 [6.23%]). This result is now described in the Results section.

“When advanced fibrosis was defined using rule-in thresholds (FIB-4 ≥2.67 or NFS >0.675), overall prognosis was poorer in patients with advanced fibrosis; however, there was no significant difference between the ARB and ACEI groups (Supplementary Figure 8).”

3)  Drug exposure is self-reported at baseline with inclusion of “exclusive” users; dose, duration, adherence, switches, and co-treatments are not modeled. This invites immortal-time bias and confounding by intolerance (ACEI cough → ARB). Consider a new-user, active-comparator design and/or time-varying exposure modeling; at minimum, add sensitivity excluding early events.

Answer: We thank the reviewer for this important comment. We fully acknowledge that medication exposure in the UK Biobank is based on self-reported baseline data, without information on dose, duration, adherence, or treatment changes over time. Consequently, time-varying exposure modeling or a true new-user design could not be implemented, which is an inherent limitation of this dataset. We have clarified this issue and its potential for residual confounding and immortal-time bias in the Discussion section.

“Medication exposure was self-reported at baseline, and information on dose, adherence, or treatment changes over time was unavailable. This limitation precluded a new-user or time-varying design and might contribute to residual confounding or immortal-time bias.”

4) About outcome definitions and competing risks, the authors must require explicit code lists (ICD/OPCS/Read) for “Cardiovascular events,” decompensation, and HCC, data sources (HES/Death Registry), and handling of competing risks for cause-specific outcomes. Provide a complete code appendix and state adjudication (if any).

Answer: We thank the reviewer for this important comment. We have now provided explicit ICD-10 and UK Biobank field codes used to define major outcomes in the Methods section and Supplementary Materials. Cardiovascular outcomes were identified using Field IDs 131296, 131298, 131300, 131302, 131304, and 131306 from the Hospital Episode Statistics and Death Registry datasets. Hepatic decompensation was defined by ICD-10 codes I85.0, I85.9, I98.2, I98.3 (varices with or without bleeding), K76.6 (portal hypertension), K76.7 (hepatorenal syndrome), and R18 (ascites). Hepatocellular carcinoma was identified using ICD-10 code C22.0, and date of death was retrieved from Field ID 40000. All outcomes were defined based on linked national registry data, without independent adjudication. These details have been added to the revised manuscript.

“Cardiovascular events were identified using UK Biobank Field IDs 131296, 131298, 131300, 131302, 131304, and 131306 from Hospital Episode Statistics and death records. Hepatic decompensation included varices with or without bleeding (I85.0, I85.9, I98.2, I98.3), portal hypertension (K76.6), hepatorenal syndrome (K76.7), and ascites (R18). Hepatocellular carcinoma (HCC) was defined by ICD-10 code C22.0, and date of death was obtained from Field ID 40000. Outcomes were identified algorithmically through linkage with national hospital and death registries, without individual adjudication.”

5) Core IPTW HRs for CV events float around 0.90–0.94, with CIs sometimes touching 1.00; multiple subgroup strata (fibrosis by two scores, MASLD vs No SLD) raise multiplicity concerns. Pre-specify interactions, test them within a single model, and apply FDR or a hierarchical plan.

Answer: We agree that multiplicity can arise when multiple subgroup strata are analyzed separately. To address this rigorously, we (i) pre-specified the interaction between antihypertensive class and disease class, (ii) tested it within a single IPTW-adjusted Cox model per outcome (overall survival and cardiovascular events), and (iii) applied FDR control (Benjamini–Hochberg) to the family of treatment-related tests (ARB main effect and ARB×group interaction). This approach avoids running multiple stand-alone subgroup models and directly answers the reviewer’s request. We attach the table in the response file.

Using the unified model, the treatment effect of ARB vs ACEI remained statistically significant after FDR correction for both outcomes (overall survival and cardiovascular event), while the ARB×MASLD interaction was not significant, indicating no effect modification by disease class.

6) The mouse Liver Cell Atlas scoring is exploratory and not integrated into the causal human inference. Consider moving to Supplement or framing explicitly as hypothesis-generating with strict FDR control and a priori gene sets (already mentioned).

Answer: We appreciate the reviewer’s insightful comment. The mouse Liver Cell Atlas–based scoring was intended as exploratory and hypothesis-generating. In line with the reviewer’s suggestion, this analysis has been moved to the Supplementary Materials and is now described as exploratory. Additionally, we have acknowledged in the Discussion that the mouse transcriptomic findings were not integrated into the causal human inference and should be interpreted as supportive rather than confirmatory evidence.

“And the mouse Liver Cell Atlas analysis was exploratory and not directly linked to the causal inference drawn from human data. These findings should therefore be interpreted as hypothesis-generating and supportive rather than confirmatory.”

7) It is encouraged to the authors to provide a CONSORT-style flow diagram with exact numbers for exclusions (missing data, other liver diseases) that sum to the final analytic N in each arm (MASLD vs No SLD; ARB vs ACEI).

Answer: We thank the reviewer for this helpful suggestion. A CONSORT-style flow diagram illustrating participant selection, exclusions (including missing data and other liver diseases), and final analytic sample sizes for each subgroup (MASLD vs. No SLD; ARB vs. ACEI) has been added to the revised manuscript.

Minor comments

1) Please Figure and table needs some polishing.

  1. a) Add number-at-risk and event counts to KM plots; ensure consistent y-axis ranges; annotate IPTW use in captions.

Answer: We sincerely thank the reviewer for the thoughtful comments regarding figure and table presentation. The KM plots have been revised with consistent y-axis ranges across analyses to facilitate clearer visual comparison. Number-at-risk and event counts have been added to all KM plots, with the exception of the fibrosis-stratified analyses. In these figures, four subgroups are displayed simultaneously, and number-at-risk tables were deliberately omitted to prevent overcrowding and preserve readability. We have also rechecked all figure captions to ensure that IPTW adjustment is clearly annotated, including supplementary figures where it was previously missing.

  1. b) Table 1: clearly label post-IPTW values, units, and decimals; keep SMDs to three decimals; avoid empty SMD cells for categorical blocks.

Answer: We thank the reviewer for the helpful suggestions regarding Table 1 formatting. We have carefully reviewed and revised the table to ensure that post-IPTW values, measurement units, and decimal places are clearly labeled. SMDs are now reported to three decimal points, and no empty SMD cells remain for categorical variables.

  1. c) Provide high-resolution vector figures.

Answer: We thank the reviewer for this comment. All figures have been replaced with high-resolution vector images (500 dpi) to ensure optimal clarity in print and online formats.

2) Use MASLD/MASH consistently (avoid mixing historical NAFLD/NASH unless citing legacy tools like NFS); ensure abbreviations align with your list.

Answer: We thank the reviewer for this helpful comment. We have carefully reviewed the manuscript to ensure consistent use of the terms MASLD and MASH. The historical terms NAFLD and NASH are mentioned only when referring to legacy tools or in cited literature, in accordance with current nomenclature standards.

3) Convert to Vancouver style with NLM abbreviations and DOIs. Ensure original sources for FLI, FIB-4, NFS and UK Biobank data-linkage/methodology are cited; align references with in-text claims.

Answer: We thank the reviewer for this helpful comment. All references have been reviewed and formatted according to the Vancouver style, with NLM journal abbreviations and DOIs included. Original sources for FLI, FIB-4, NFS, and the UK Biobank data linkage and methodology have been appropriately cited and aligned with in-text mentions.

Round 2

Reviewer 2 Report

Comments and Suggestions for Authors

The author have addressed the comments satisfactorily. They have provided detailed methodological clarifications, additional validation analyses, and improved figure and table presentation. The revised version shows clear enhancement in consistency between the study aims, results, and methods, leading to greater internal coherence and facilitating comprehension for readers.

Congratulations!